# Comment on Geoffroy, M.-C.; de Thé, H. Classic and Variants APLs, as Viewed from a Therapy Response. *Cancers* 2020, *12*, 967

**DOI:** 10.3390/cancers13235883

**Published:** 2021-11-23

**Authors:** Zhan Su, Xin Liu

**Affiliations:** 1Department of Haematology, The Affiliated Hospital of Qingdao University, Qingdao 266000, China; 2Department of Stem Cell Transplantation, Blood Diseases Hospital & Institute of Hematology, Chinese Academy of Medical Sciences & Peking Union Medical College, Tianjin 300020, China; liuxin@ihcams.ac.cn

Acute promyelocytic leukemia (APL) is a unique and very deeply studied acute myeloid leukemia. The typical molecular genetic feature of APL is the rearrangement of RARA gene with PML and a few with other partner genes. All-trans retinoic acids and arsenic trioxides are effective agents against APL harboring *PML-RARA*. However, there are still some rare cases lacking *RARA* fusion although they share the same morphology and immunocytochemistry features with typical APL. The classification of such cases remains controversial. They are usually referred to as acute promyelocytic-like leukemias (APLL), and sometimes they were morphologically classified as M3 according to the FAB classification. They are not regarded as APL in the WHO’s classification of myeloid neoplasms and acute leukemia. For years, the underlying molecular pathology of APL-like leukemia has remained concealed. Some researchers supposed that APL-like leukemia might bear genetic mutation of other members of retinoic acid receptors (RARs), such as RARB or RARG [1].

Due to the limitations of technology, identification of unknown fusion genes has been a relatively difficult task. In recent years, the emerging high-throughput sequencing technologies have provided powerful research tools for the discovery of novel fusion genes. In 2011, a novel *NUP98-RARG* fusion gene was discovered by Such E et al. in a Spanish case of APL-like leukemia, and it was the first reported fusion gene of this kind of leukemia. Since 2017, several other novel fusion genes harboring RARG have been found. Geoffroy et al. made a relatively comprehensive review in this field. In this review, a total of four RARG-related fusion genes are summarized, namely *CPSF6-RARG* & *RARG-CPSF6*, *NUP98-RARG*, *PML-RARG,* and *NPM1-RARG-NPM1*. *NUP98-RARG* accounts for a total of three cases and *CPSF6-RARG* & *RARG-CPSF6* for five cases. *CPSF6-RARG* & *RARG-CPSF6* own the most variants, consisting of four types of *CPSF6-RARG* and one type of *RARG-CPSF6* [2]. In fact, another patient with *CPSF6-RARG* has been reported in 2019 [3]. Since then, two more cases of *CPSF6-RARG* and two cases of *NUP98-RARG* have been reported [4,5,6,7]. Therefore, the number of reported patients has reached eight cases for *CPSF6-RARG* and six for *NUP98-RARG*. Their biological and clinical information are summarized in Table 1 and Table 2, respectively.

Actually, RARG-related fusion genes that have been identified were not merely the four types mentioned. Su et al., reported a 43-year-old man with leukocytosis, anemia, and thrombocytopenia. Similarly, the morphology and immunological characteristics of the blasts were the same as APL. Transcriptome sequencing identified a new *HNRNPC-RARG* fusion gene and its reciprocal. The patient received ATRA induction therapy for three weeks, but the bone marrow blasts were almost unchanged. Then, combined chemotherapy (homoharringtonine, daunorubicin and cytarabine, HAD) was applied and complete remission was achieved. After that, he was given several courses of consolidation and intensive chemotherapy consecutively, although he relapsed one year later. Re-induction therapy medications included ATRA, arsenic trioxide, and combination chemotherapy, which, as inu other such cases, have not worked [16].

The *RARG* gene is located at chromosome 12q13. Several more APL-like cases with chromosome translocation involving 12q13 have been reported in previous pieces of literature, but further molecular detections were not performed (Table 3). Due to the fact that this kind of disease is relatively rare and due to the current limitations of research conditions, there may be more such cases unpublished. Actually, we also know that this situation does exist. As exhibited in Table 3, most cases are t(11;12) translocations, which are consistent with the location of the *NUP98-RARG* fusion gene. One of them is t(4;12) translocation, suggesting that there may be undiscovered new fusion genes. These cases also did not respond to the treatment of retinoic acid or arsenic trioxide.

Interestingly, there is another group of AML with t(11;12)(p15;q13), of which the cell morphology is not promyelocyte type, but classified into M1, M2, M4, and M5 according to FAB classification. In the fusion gene of such cases, one partner gene is also *NUP98* located on chromosome 11p15, and the other is one of *HOXC* (*HOXC11*, *HOXC13*) gene clusters [18,21]. It should be noted that a series of genes are densely distributed in the region of human chromosome 12q13, including *RARG* and *HOXC*.

All reported cases of APLL with 12q13 rearrangement achieved no response to ATRA or arsenic trioxide treatment in vivo. In vitro culture of bone marrow blast cells from the first patient with *NUP98-RARG* also confirmed the failure, although hematopoietic stem cells transfected with *NUP98-RARG* were demonstrated to be sensitive to ATRA. Some researchers attributed this contradiction to the influence of the in vitro culture environment or additional mutations [22]. In fact, a series of RARs modulators have been synthesized, some of which are pan-antagonists/agonists and some of which are selective ones. ATRA is one of pan- agonists [23]. Researchers have observed that BMS961 (an RARG agonist) has not shown any effect on leukemia cells with *RARG-CPSF6* cultured in vitro [10].

In most of the published reports and WHO classification, as mentioned earlier, APLL is regarded more as a subclass of AML than as an APL. However, the authors of this review pointed out in another article [24] that RARB and RARG are also retinoic acid receptors and play a key role in leukemogenesis through their rearrangement (similar to RARA). Moreover, APLL carrying *RARG-CPSF6* shared the same gene expression profiles with classical APL harboring *PML-RARA* [10]. Although APLL is not responsive to ATRA, new target drugs (e.g., other RARs modulators) may be explored to improve the outcome of APLL patients. Thus, we think it seems to be reasonable to classify the APL and APLL into a same subclass and the APL with RARs rearrangements.

However, there have existed cases of APLL lacking RARs fusion genes. Researchers further explored other possible molecular abnormalities (e.g., dysregulation of *RARA* and *RARG* has been reported) [25,26]. Zhao et al., identified two new fusion genes *NPM1-CCDC28A* and *TBC1D15-RAB21* in two cases of children without RARs rearrangement. The abnormal chromosome of the third patient was t(1;11)(q21;q23), and DNA sequencing identified two fusion genes, *KMT2A-MLLT11* and *RPRD2-KMT2A* [3]. In addition, sporadic APLL literature could be retrieved which did not carry out in-depth study of molecular pathology, and some involved known genetic aberration such as JAK2 V617F mutation [27].

In conclusion, the identification of *RARG*/*RARB* related fusion genes over the last decade has revealed the mystery of APLL, and it can be predicted that more cases will be reported in the future. However, due to the insensitivity to ATRA and arsenite acid, there is still a lack of effective targeted drugs. In addition, there are some cases lacking RARs rearrangements, of which the molecular pathology remains to be clarified.

## Figures and Tables

**Table 1 cancers-13-05883-t001:** Cases of APLL harboring *CPSF6-RARG* or *RARG-CPSF6*.

No.	Age/ Gender	Promyelocytes Percentage	Cytogenetics and Molecular Aberrations	Immunophenotyping	ATRA/ATO Responsivity	Treatment and Efficacy	References
1	48/F	89%	92, XXXX (2); *CPSF6-RARG*; *DNMT3A*-G587fs mutation	Positive: CD13, CD33, and MPO; partially positive: CD9, CD64; negative: HLA-DR, CD117, CD34, CD14 and CD11b	ATRA+ATO, none	ATRA+ATO+idarubicin, ATRA+ATO+IA+G-CSF, then decitabine, NR. Abondoned treatment then died	[8]
2	51/F	87.5%	del(12)(p12)(2)/46,XX (18); *CPSF6-RARG*; *WT1* and *K-RAS* mutations	Positive: CD13, CD33, MPO, and CD9; partially positive: CD34; negative: HLA-DR, CD2, CD7, CD10, CD11c, CD14, and CD38	ATRA, none	ATRA+daunorubicin, NR; then DA, morphologic remission. followed by 2 courses of HD-AC and 2 courses of 7+3 chemotherapy, CR	[8]
3	38/M	65%	46,XY (20); *CPSF6-RARG*; *WT1*-R462Q mutations	Positive: CD117, CD123, CD34, CD33 and CD13; partially positive: CD9, CD64; negative: CD11b, HLA-DR, CD38, CD56, and CD14	ATRA, none. suspected differentiation syndrome	ATRA+RIF, then MA as induction therapy, died on the 37th day	[9]
4	26/M	60% blasts, 15% promyelocytes	45,X,-Y (10)/45, idem, add(6)(q?13)(2)/46,XY (8); *RARG-CPSF6*; *BMPR1A*, *NEAT1*, *WT1* mutations	Positive: CD33, CD13, CD64; partially positive: CD117; weak: HLA-DR; negative: CD34, CD56, CD19, CD2, CD5, CD123, CD14, CD11b, and TdT	ATRA, not exhibited	ATRA+IA, lacking efficacy introduction	[10]
5	5/M	Not exhibited	46,XY; *RARG-CPSF6*/*CPSF6-RARG*	Not exhibited	Not exhibited	ATRA+chemotherapy, relapse then death(11 months from diagnosis)	[3]
6	55/M	93%	46,XY; *RARG-CPSF6*	Positive: CD13, CD33, CD117, CD56; negative: HLA-DR, CD34, CD38, CD15, CD14, CD7, CD2, CD3, CD4, CD8, CD19, CD20, CD10	ATRA+ATO, none	ATRA+ATO, switched to IA, NR; then HA, CR. followed by 4 courses of HA and 1 course of EA as maintenance therapy	[11]
7	67/F	72%	46,XX; *CPSF6-RARG*; *WT1* mutation	Positive: MPO, CD13, CD33; partially positive: CD71, negative: CD14, CD19, CD34, CD38, CD64, CD117, CD11b, CD11c, HLA-DR	ATRA(10 days), NR	1 week of ATRA then plus HA, died during the course	[6]
8	55/M	5.5% blasts, 88% promyelocytes	46,XY; *CPSF6-RARG*	Positive: CD33, CD13, CD117, CD56, negative: CD34, HLA-DR	ATRA, none	ATRA plus 3 + 7 schedule, NR; then HA, sustained CR	[7]

No., number; ATRA, all-trans retinoic acid; ATO, arsenic trioxide; M, male; F, female; RIF, oral arsenic realgar-indigo naturalis formula; IA, idarubicin and cytarabine; DA, Daunorubicin and cytarabine; HD-AC, high dose cytarabine; MA, mitoxantrone and cytarabine; HA, homoharringtonine and cytarabine; EA, etoposide and cytarabine; NR, no response; CR, complete response.

**Table 2 cancers-13-05883-t002:** Cases of APLL harboring *NUP98-RARG*.

No.	Age/ Gender	Promyelocytes Percentage	Cytogenetics and Mutation	Immunophenotyping	ATRA/ATO Responsivity	Treatment and Efficacy	References
1	35/M	80%	46,XY,t(11;12)(p15;q13) (16)/46,XY (4)	Positive: CD13, CD33, CD45, CD117, cMPO; weakly positive: CD34; negative: HLA-DR, B or T-cell markers	ATRA, unknown in vivo, none in vitro	IA, CR, followed by consolidation chemotherapy, then auto-PBSCT, CR for 2 years. Died when relapsed during ATRA+salvage treatment+UCBT	[12,13]
2	45/F	94.5%	46,XX,t(11;12)(p15;q13) (16)/46 XX (4); *WT1*-R445W mutation	Positive: MPO, CD117, CD33, CD13, CD38, CD64; negative: CD34, CD11b, HLA-DR, CD56, CD14, B or T-cell markers	None	ATRA+ATO, switched to CA, NR and died	[14]
3	22/M	91%	46,XY,t(11;12)(p15;q13); *WT1*(c.1255_1256insGG, c643C>T) mutations	Positive: CD117, CD13, CD33, partially positive: HLA-DR; negative: CD34	None	ATRA+ATO+ idarubicin, then HAA, NR; switched to DA, PR; followed by DA, CR	[15]
4	47/F	96.5%	45,X,–X, del(9)(q13q22), t(11;12)(p15;q13) (20); *IDH2*, *TET2*, *ASXL1*, *TP53*, *WT1*(exon7, exon9) mutations	Positive: MPO, CD13, CD33, HLA-DR, CD56	ATRA+ATO (14 days), none	ATRA+ATO, IA, HIAG, sequentially, CR; then HIAG, 2 cycles of half-CAG, 2 cycles of HA as consolidation chemotherapy; leukemia-free in 24-month follow-up	[4]
5	18/M	90.5%	46,XY; *RUNX1*:c.319C>A mutation(onset), 8 point mutations in *WT1*(relapse)	Positive: CD117, CD13, CD33, CD9, CD64, CD123, cMPO; negative: HLA-DR, CD34, CD38, CD11b, B or T-cell markers	ATRA (5 days)+ ATO (13 days), NR	ATRA+ATO, switched to DA, NR; then HAA, PR; HAA, HD-AC, CR; the patient abandoned further treatment and relapsed 3 months later and died	[5]
6	33/M	93%	46,XY	Positive: CD33, CD13, CD117; negative: CD34, HLA-DR	None	ATRA plus 3 + 7 schedule, NR and died	[7]

No., number; ATRA, all-trans retinoic acid; ATO, arsenic trioxide; M, male; F, female; auto-PBSCT, autologous peripheral blood stem-cell transplant; IA, idarubicin and cytarabine; UCBT, umbilical cord blood transplantation; CA, aclamycin and cytarabine; HAA, homoharringtonine, aclarubicin, and cytarabine; DA, daunorubicin and cytarabine; HIAG, homoharringtonine, idarubicin, cytarabine, and granulocyte colony-stimulating factor(GCSF); half-CAG, half-dose aclarubicin, cytarabine, and GCSF; HA, homoharringtonine and cytarabine; HD-AC, high dose cytarabine; NR, no response; CR, complete response; PR, partial response.

**Table 3 cancers-13-05883-t003:** Cases of APLL with 12q13 rearrangement and unidentified fusion gene.

No.	Age/ Gender	Promyelocytes Percentage	Cytogenetics and Mutation	Immunophenotyping	ATRA/ATO Responsivity	Treatment and Efficacy	References
1	37/M	87%	46,XY,t(4;12)(q11;q13) (29)/46,XY (6)	Positive: CD13, CD33, negative: CD34, HLA-DR	ATO (15 days), NR	ATO+CAG as induction therapy, died during the course	[17]
2	14/M	90%	46,XY,t(11;12)(p15;q13)	Positive: CD13, CD33, CD 14, CD 64, CD34	ATRA (2 weeks), NR; ATO (1 month), none	ATRA followed by ATO, then DA, MA as induction therapy, NR and died	[18]
3	51/M	81.5%	48,X Y,t(11;12) (p15;q13),+14,+21 (10)/46,XY (10); *FLT3-T K D* mutation	Positive: CD13, CD33; partially positive: CD4, CD15, CD34, CD38, CD64, CD117; minority positive: CD11b; negative: HLA-DR	ATRA (8 days), NR, ATO (21 days), none	ATRA+ATO, then HDA as induction therapy, CR; followed by D+ID-AC, M+ID-AC, HA, HA, MA, MA as consolidation therapy	[18]
4	21/M	89%	46,XY,t(11;12)(p15;q13) (20)	Positive: CD117, CD33, CD13; partially positive: CD64; negative: CD34, CD14, CD56, CD4, CD19, HLA-DR	None	ATRA+ATO+idarubicin, HAA, NR; then IA, CR; died after consolidation therapy	[19]
5	60/M	85.6%	46,XY,t(11;12)(p15;q13)	Positive: cMPO, CD13, CD33, CD117, CD64, negative: CD34, HLA-DR	None	ATRA+ATO, NR; then DA, CR	[20]
6	49/M	78%	46,XY,t(11;12)(p15;q13)	Positive: cMPO, CD13, CD33, CD117, CD64, negative: CD34, HLA-DR	None	ATRA+ATO, NR; then HAA, CR	[20]

No., number; ATRA, all-trans retinoic acid; ATO, arsenic trioxide; M, male; F, female; CAG, aclamycin, cytosine arabinoside, and granulocyte-colony stimulating factor(G-CSF); DA, daunorubicin and cytarabine; MA, mitoxantrone and cytarabine; NR, no response; CR, complete response; HDA, homoharringtonine, daunorubicin and cytarabine; D+ID-AC, daunorubicin and intermediate dose cytarabine; M+ID-AC, homoharringtonine and intermediate dose cytarabine; HA, homoharringtonine and cytarabine; HAA, homoharringtonine, aclamycin, and cytarabine; IA, idarubicin and cytarabine.

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
