# Peer review of "Comment on Geoffroy, M.-C.; de Thé, H. Classic and Variants APLs, as Viewed from a Therapy Response. *Cancers* 2020, *12*, 967"

_cancers, 2021, doi:10.3390/cancers13235883_

Round 1
Reviewer 1 Report
In this Comment on Geoffroy, M.-C.; et al. (Cancers 2020, 12, 3 967), the authors, Zhan Su and Xin Liu, make an update of the literature on cytogenetic alterations associated with acute promyelocytic-like leukemias. However, some inaccuracies and/or oversights have been found in the manuscript that need to be improved.
- Relatively to CPSF6-RARG and NUP98-RARG alterations, the most recent bibliography should be cited.
- Jiang M, Zhou YR, Zhan Y, Zhang HQ, Zhang Q, Guo Y, Zhang ZL. [Application of transcriptome sequencing and fusion genes analysis in the diagnosis of myeloid leukemia with normal karyotype]. Zhonghua Yi Xue Za Zhi. 2021 Apr 6;101(13):939-944. Chinese. doi: 10.3760/cma.j.cn112137-20201103-03005. PMID: 33789375.
- Han X, Jin C, Zheng G, Li Y, Wang Y, Zhang E, Zhu H, Cai Z. Acute myeloid leukemia with CPSF6-RARG fusion resembling acute promyelocytic leukemia with extramedullary infiltration. Ther Adv Hematol. 2021 Jan 9;12:2040620720976984. doi: 10.1177/2040620720976984. PMID: 33473264; PMCID: PMC7797573.
- Therefore, the number of reported patients would achieve 8 cases for CPSF6-RARG and 6 for NUP98-RARG. Since biological and clinical information are described in most of the case reports, it would be interesting to include two additional tables, one for each alteration, following similar structure of Table 1 in the manuscript, in order to facilitate the comparison among cases.
- Sentences in lines 20 and 23 should be endorsed by appropriate citations.
- Authors commented that Zhao, J et al. 2020 did not describe details of clinical and molecular features for the reported CPSF6-RARG patient (in lines 36-37). We kindly disagree. Although limited, some additional information can be found in figure 3b, Supplementary table 9 and the main text. It is important to note that this article had already been cited by Geoffroy, MC et al. 2020, thus it cannot be considered as a “new” case.
- In order to make the manuscript more interesting, we propose to change the order of the paragraphs, describing the alterations/cases according to their frequency: CPSF6-RARG, NUP98-RARG, HNRNPC-RARG, cases with t(11;12) translocations, etc.
- In most of the published reports, atypical APL is classified more as a subclass of AML than as an APL, mainly due to the observed mutational profiles and response to treatments. A paragraph speculating on this point would be valuable to the readers.
Minor issues:
- In lines 13-14, “The typical genetic feature of APL is the rearrangement of RARA gene with PML,” should be preferred to “The typical molecular genetic feature of APL is the rearrangement of RARA gene, mainly with PML”.
- In lines 14-15, “All-trans retinoic acid” should be preferred to “Retinoic acid” as well as “arsenic trioxide” instead of “Arsenious acid”.
- In line 18, a space should be inserted between “leukemias” and “(APLL)”.
- In line 19, “M2” should be replaced by “M3”.
- In line 20, “WHO classification of myeloid neoplasms and acute leukemia” should be preferred to “WHO classification of tumors either”.
- In line 25, “sequencing” should be added between “high-throughput” and “technologies”.
- In lines 32-33, “Wherein NUP98-RARG accounts for a total of three cases and CPSF6-RARG & RARG-CPSF6 for 5 cases” should replace current sentence.
- In line 43, according to Su Z. et al. 2020, “45-year-old man” should be corrected with “43-year-old man”.
- In line 53, please replace “locats” with “is located”.
Author Response
Point 1: Relatively to CPSF6-RARG and NUP98-RARG alterations, the most recent bibliography should be cited.
- Jiang M, Zhou YR, Zhan Y, Zhang HQ, Zhang Q, Guo Y, Zhang ZL. [Application of transcriptome sequencing and fusion genes analysis in the diagnosis of myeloid leukemia with normal karyotype]. Zhonghua Yi Xue Za Zhi. 2021 Apr 6;101(13):939-944. Chinese. doi: 10.3760/cma.j.cn112137-20201103-03005. PMID: 33789375.
- Han X, Jin C, Zheng G, Li Y, Wang Y, Zhang E, Zhu H, Cai Z. Acute myeloid leukemia with CPSF6-RARG fusion resembling acute promyelocytic leukemia with extramedullary infiltration. Ther Adv Hematol. 2021 Jan 9;12:2040620720976984. doi: 10.1177/2040620720976984. PMID: 33473264; PMCID: PMC7797573.
Response 1: We are very grateful to the reviewer. According to the reviewers' reminders, we have added these two references to our comment.
In addition, we have recently discovered a consultation post on the Internet (https://www.haodf.com/kanbing/6480749833.html), which introduced a child with APLL, and a RARG-HNRNPC fusiong gene was detected during the course of diagnosis. As far as we know this may be the second case with HNRNPC-RARG harboring APLL. But the case is not officially published, and therefore it seems that we have no opportunity to introduce the case in the comment.
Point 2:Therefore, the number of reported patients would achieve 8 cases for CPSF6-RARG and 6 for NUP98-RARG. Since biological and clinical information are described in most of the case reports, it would be interesting to include two additional tables, one for each alteration, following similar structure of Table 1 in the manuscript, in order to facilitate the comparison among cases.
Response 2: Thank the reviewer, we have added these tables following your advice.
Point 3: Sentences in lines 20 and 23 should be endorsed by appropriate citations.
Response 3: Thank the reviewer, we have added an article.
Point 4: Authors commented that Zhao, J et al. 2020 did not describe details of clinical and molecular features for the reported CPSF6-RARG patient (in lines 36-37). We kindly disagree. Although limited, some additional information can be found in figure 3b, Supplementary table 9 and the main text. It is important to note that this article had already been cited by Geoffroy, MC et al. 2020,thus it cannot be considered as a “new” case.
Response 4: Thank the reviewer for carefully reviewing our comments. It is really true as Reviewer suggested. Article of Zhao, J et al. 2019 (not 2020) cited by Geoffroy, MC et al. 2020,but Geoffroy, MC et al. 2020 only focused on the classic APL cases in Zhao, J et al. 2019 (cited in Unit 3), the patient carrying CPSF6-RARG was missed (should have been cited in Unit 4.2.2. and Table 1.). If this case has been added, there are a total of 6 patients CPSF6-RARG in the review, rather than 5 summarized in table 1..
In addition, we found that the reference [5] in our comment had appeared in the review of Geoffroy, MC (reference 85), so we deleted it.
Point 5: In order to make the manuscript more interesting, we propose to change the order of the paragraphs, describing the alterations/cases according to their frequency: CPSF6-RARG, NUP98-RARG, HNRNPC-RARG, cases with t(11;12) translocations, etc.
Response 5: Thank the reviewer. Considering the Reviewer’s suggestion, we adjusted it in this order.
Point 6: In most of the published reports, atypical APL is classified more as a subclass of AML than as an APL, mainly due to the observed mutational profiles and response to treatments. A paragraph speculating on this point would be valuable to the readers.
Response 6: Thank the reviewer, we have added a paragraph before the penultimate paragraph.
Minor issues:
- Point 1: In lines 13-14, “The typical genetic feature of APL is the rearrangement of RARA gene with PML,” should be preferred to “The typical molecular genetic feature of APL is the rearrangement of RARA gene, mainly with PML”.
Response 1: Thank the reviewer, we have deleted the word.
- Point 2: In lines 14-15, “All-trans retinoic acid” should be preferred to “Retinoic acid” as well as “arsenic trioxide” instead of “Arsenious acid”.
Response 2: Thank the reviewer, we have changed the drug name.
- Point 3: In line 18, a space should be inserted between “leukemias” and “(APLL)”.
Response 3: We are very sorry for our negligence, a space have been inserted.
- Point 4: In line 19, “M2” should be replaced by “M3”.
Response 4: Thanks,we have replaced it.
- Point 5: In line 20, “WHO classification of myeloid neoplasms and acute leukemia” should be preferred to “WHO classification of tumors either”.
Response 5: Thank the reviewer, we have amended the phrase.
- Point 6: In line 25, “sequencing” should be added between “high-throughput” and “technologies”.
Response 6: Thank the reviewer, we have added it.
- Point 7: In lines 32-33, “Wherein NUP98-RARG accounts for a total of three cases and CPSF6-RARG & RARG-CPSF6 for 5 cases” should replace current sentence.
Response 7: Thanks, we have replaced the sentence according to reviewer’s direction.
- Point 8: In line 43, according to Su Z. et al. 2020, “45-year-old man” should be corrected with “43-year-old man”.
Response 8: Thank the reviewer, we are very sorry for our negligence, and we have made it right.
- Point 9: In line 53, please replace “locats” with “is located”.
Response 9: Thanks, we are very sorry for our incorrect writing, and we have corrected it.
Special thanks to you for your good comments.
Reviewer 2 Report
The authors Zahn Su and Xin Liu comment on a straight forward written, quite comprehensive Review of Geoffroy et al., originally submitted in 2020. As the Review is informative and significant for both ends, molecular understanding of the disease and clinical actionability by targeted treatment, the comment is of significant importance as well as new evidence emerged since then. The authors update on more genomic alterations in non PML-RARA harboring APL. Further, additional evidence indicates a new variant of CPSF6-RARG fusion gene. From an oncologists point of view careful decision making on a molecular rational seems to be even more important as patients with those newly described APLL did not achieve responses to ATRA or arsenic trioxide.
Author Response
Point 1:The authors Zhan Su and Xin Liu comment on a straight forward written, quite comprehensive Review of Geoffroy et al., originally submitted in 2020. As the Review is informative and significant for both ends, molecular understanding of the disease and clinical actionability by targeted treatment, the comment is of significant importance as well as new evidence emerged since then. The authors update on more genomic alterations in non PML-RARA harboring APL. Further, additional evidence indicates a new variant of CPSF6-RARG fusion gene. From an oncologists point of view careful decision making on a molecular rational seems to be even more important as patients with those newly described APLL did not achieve responses to ATRA or arsenic trioxide.
Response 1:
We are very grateful to the reviewers for their guidance and encouragement to our work. At present, the classification of hematological malignancies pays more attention to molecular abnormalities, and the corresponding targeted drugs are developed. Although emerging one after another, new targeted drugs are far from reaching the efficacy of treatment for APL and CML. This suggests that the oncogenesis of most blood tumors is much more complex than classical APL and may not manly depend on a single molecular abnormality. The discovery of fusion genes involving RARG/RARB in APLL has solved the mystery of APLL's driving molecular mutation that has been pending for many years. However, its no response to retinoic acid does not meet people's expectations. Nevertheless, the expectation taken for granted comes from the assumption of PML-RARA. In fact, the characteristics of RARG/RARB fusion gene are hardly understood. Although RARA, RARB and RARG have high sequence homology and similar functions, they also have their own particular features. Accordingly, there are a series of compounds as agonists/antagonists to retinoic acid receptors, including both pan-agonists/antagonists and selective ones. As mentioned in our comment, it might be considered to to try these compounds before classifying APLL as a multi-driver-mutation carcinogenic malignancy.
Thanks again to the reviewers!
Round 2
Reviewer 1 Report
-